# Diffusion-Weighted MRI in the Genitourinary System

**DOI:** 10.3390/jcm11071921

**Published:** 2022-03-30

**Authors:** Thomas De Perrot, Christine Sadjo Zoua, Carl G. Glessgen, Diomidis Botsikas, Lena Berchtold, Rares Salomir, Sophie De Seigneux, Harriet C. Thoeny, Jean-Paul Vallée

**Affiliations:** 1Division of Radiology, Geneva University Hospitals and University of Geneva, 1205 Geneva, Switzerland; christine.sadjo@hcuge.ch (C.S.Z.); carl.glessgen@hcuge.ch (C.G.G.); diomidis.botsikas@hcuge.ch (D.B.); raresvincent.salomir@hcuge.ch (R.S.); jean-paul.vallee@hcuge.ch (J.-P.V.); 2Division of Nephrology, Geneva University Hospitals, 1205 Geneva, Switzerland; lena.berchtold@hcuge.ch (L.B.); sophie.deseigneux@hcuge.ch (S.D.S.); 3Division of Radiology, Hôpital Cantonal Fribourgois, 1752 Villars-sur-Glâne, Switzerland; harriet.thoeny@h-fr.ch

**Keywords:** genitourinary MRI, diffusion, prostate, kidney, female pelvis, cancer

## Abstract

Diffusion weighted imaging (DWI) constitutes a major functional parameter performed in Magnetic Resonance Imaging (MRI). The DW sequence is performed by acquiring a set of native images described by their b-values, each b-value representing the strength of the diffusion MR gradients specific to that sequence. By fitting the data with models describing the motion of water in tissue, an apparent diffusion coefficient (ADC) map is built and allows the assessment of water mobility inside the tissue. The high cellularity of tumors restricts the water diffusion and decreases the value of ADC within tumors, which makes them appear hypointense on ADC maps. The role of this sequence now largely exceeds its first clinical apparitions in neuroimaging, whereby the method helped diagnose the early phases of cerebral ischemic stroke. The applications extend to whole-body imaging for both neoplastic and non-neoplastic diseases. This review emphasizes the integration of DWI in the genitourinary system imaging by outlining the sequence’s usage in female pelvis, prostate, bladder, penis, testis and kidney MRI. In gynecologic imaging, DWI is an essential sequence for the characterization of cervix tumors and endometrial carcinomas, as well as to differentiate between leiomyosarcoma and benign leiomyoma of the uterus. In ovarian epithelial neoplasms, DWI provides key information for the characterization of solid components in heterogeneous complex ovarian masses. In prostate imaging, DWI became an essential part of multi-parametric Magnetic Resonance Imaging (mpMRI) to detect prostate cancer. The Prostate Imaging–Reporting and Data System (PI-RADS) scoring the probability of significant prostate tumors has significantly contributed to this success. Its contribution has established mpMRI as a mandatory examination for the planning of prostate biopsies and radical prostatectomy. Following a similar approach, DWI was included in multiparametric protocols for the bladder and the testis. In renal imaging, DWI is not able to robustly differentiate between malignant and benign renal tumors but may be helpful to characterize tumor subtypes, including clear-cell and non-clear-cell renal carcinomas or low-fat angiomyolipomas. One of the most promising developments of renal DWI is the estimation of renal fibrosis in chronic kidney disease (CKD) patients. In conclusion, DWI constitutes a major advancement in genitourinary imaging with a central role in decision algorithms in the female pelvis and prostate cancer, now allowing promising applications in renal imaging or in the bladder and testicular mpMRI.

## 1. Introduction

In the broad range of clinical imaging methods, diffusion-weighted MR imaging (DWI) stands out for its exceptional value to patient management as well as for its fascinating technique. With a spatial resolution close to 1 mm, Diffusion-Weighted (DW) sequences probe the free motion of water molecules in the tissue at the micrometer level, with an amplification factor close to a thousand. First introduced in 1986 by Le Bihan et al. [1], DWI experienced major development after demonstration of its ability to detect cerebral ischemia long before any other non-invasive methods [2,3]. While the process of impaired water diffusion following cellular swelling is still partly understood [4], the use of DWI was rapidly extended to other diseases. As water diffusion also decreases in tumors due to their high cellular density, many successful applications of DWI have been validated in oncology and, although the initial applications were limited to the brain, DWI expanded rapidly to other body parts including the genitourinary system.

The genitourinary system is usually investigated by ultrasound or axial computed tomography (CT) as first line imaging modalities to detect signs of malignant lesions or to perform disease staging. Yet, magnetic resonance imaging (MRI) has emerged as a key player in the diagnosis and characterization of tumorous and non-tumorous diseases, in part due to its superior tissue contrast. As a matter of fact, MRI not only grants high resolution morphological images but also provides various functional information, such as tissue oxygenation, perfusion, or diffusion. Among these functional imaging techniques, DWI certainly impacts the management of genitourinary cancer patients most. In particular, DWI has become a pivotal tool in the diagnosis and staging of many gynecologic and prostatic cancers. Finally, driven by the advancement of respiratory motion mitigation methods, DWI has also been successfully applied to renal imaging.

Beyond renal cancer, DWI appears as an emerging tool that will most likely play a major part in the clinical management of non-tumorous renal diseases. The aim of this work is to review the current applications as well as potential future use cases of DWI, with a focus on the female pelvis, the prostate, the bladder, the penis, the testis and the kidneys.

## 2. Principles of Diffusion-Weighted MRI in the Genitourinary System

Water is the most abundant molecule in soft tissues. Each water molecule bears two hydrogen nuclear spins, which are the physical source of the MRI signal in the overwhelming majority of clinical applications. Water molecules undergo chaotic perpetual microscopic motion, called molecular diffusion, exploring the available spaces in intra- and extracellular compartments. In presence of a strong static magnetic field, these hydrogen nuclear spins begin to rotate around the axis of the field in a process called precession. The precession frequency is directly proportionate to the amplitude of the static magnetic field.

The well-known spin echo MR technique [5] yields intra-voxel refocusing of spins by “time mirroring” the individual differences in precession frequencies. These frequency offsets can occur due to local inhomogeneities of the static magnetic field or can be dynamically induced by the application of magnetic gradient pulses. Spin echo refocusing is imperfect if the observed spins are undergoing chaotic motion, corresponding to a partial loss of spin coherence and attenuation of the spin echo signal intensity [6]. Therefore, the observed MRI signal contains information on the molecular motion of water and specifically, the motion restriction due to various biological structures [4].

In a free medium, the probability to locate a given water molecule after a given period of time is a 3D isotropic Gaussian function, with full width at half maximum (FWHM) increasing proportionally to the square root of the observation time. In this case, a scalar value, the apparent diffusion coefficient (ADC, mm^2^·s^−1^), is determined as a measurement of the diffusion’s magnitude [7] and the MRI signal attenuation is a single exponential function of the sequence’s gradient weighting. The strength of such magnetic gradients is named using the letter “b” followed by a numeric variable representing the amplitude and duration of the applied gradients, expressed in SI base units of s·mm^−2^. Typical pairs of b-values vary between 0–500 or 1000 s/mm^2^ for the abdomen and 0–200 and 1000 s/mm^2^ for the pelvis [8].

In prostate imaging, values range between 0 and 2000 s/mm^2^, for example b50, b500, b1000, b1500 and b2000. Diffusion-weighted sequences using gradient values higher than 1000 s/mm^2^ can be referred to as high (or even ultra-high) b-values DW sequences and their significance in prostate MRI has been demonstrated by multiple studies [9,10]. In the presence of such gradients, if barrier-like structures restrict molecular movements in a tissue, a high MR signal will be preserved and the tissue will appear distinctly hyperintense on DW images and hypointense on ADC, reflecting the reduced water diffusion. In theory, the simplest way to measure ADC only requires DWI acquisitions for two b-values and a monoexponential fit, but other more complex models have been developed to better describe the water molecule motion inside biological tissues. These models have been mainly investigated in the prostate and are described in the dedicated section.

Since MRI does not directly sample the object but its spatial frequencies (deposited in the so-called k-space) it is particularly sensitive to motion. Tissue displacement during the acquisition yields moderate and sometimes severe artifacts [11], for instance, blurring, ghosting and alteration of tissue contrast. Various mitigation techniques have been developed to correct for movement during the acquisition. The most basic method to avoid respiratory motion is to acquire images during a breath-hold. Temporal synchronization of MR signal acquisition with the physiological motion was then obtained using triggering or synchronization to the ECG or respiratory waveforms. More elaborate approaches consist of tracking the tissue position by using MR-based navigators to prospectively or retrospectively correct for motion. In a clinical setting, the acquisition of high quality renal or pelvic DW images within a single apnea is not always feasible. Therefore, motion compensation techniques may be required to improve the image quality of DWI and to avoid the confounding effect of macroscopic movements on water diffusion [12,13].

To further reduce the effect of physiologic motion, DWI is conventionally acquired using single-shot encoding schemes which are referred to as echo planar imaging (EPI). In EPI, the initial excitation RF pulse generating the MR signal is followed by a series of gradient patterns and refocusing RF pulses that cover the k-space of each individual slice. The k-space in the frequency domain is then converted into an image using a mathematical operation, the Fourier transform. EPI is prone to geometric distortion when the local magnetic field is inhomogeneous and to other more complex artifacts, such as imperfect saturation of the fat signal. One solution to overcome these limitations is the segmentation of k-space yet at the cost of an increased acquisition time. The “Resolve” (REadout Segmentation Of Long Variable Echo-trains) technique [14] consists of the shortening of read-out lines in k-space which are subdivided into several parallel bands, at least three. This feature allows a reduction of the echo time and of the frequency-encoding time. In return, the technique delivers sharper images that are generally free of distortion and of high spatial resolution, allowing for broad use in prostate and renal DWI.

## 3. Diffusion-Weighted Female Pelvis Imaging

Magnetic resonance imaging is a complementary imaging modality usually performed after an ultrasound. DWI is crucial and performed in most female pelvic studies in addition to conventional morphologic T1- and T2-Weighted (T2W) sequences, as shown in Figure 1. DWI, together with dynamic contrast-enhanced (DCE) imaging, is part of the functional imaging apparatus which in recent times increased the diagnostic performance of MRI in the field of gynecologic oncology. As DWI suffers from poor spatial resolution, and therefore, less anatomical definition, it has to be used in association with a morphologic T2W sequence [15]. DWI is particularly useful in the assessment of endometrial and cervical cancer, helping to differentiate between benign and malignant uterine or ovarian lesions and assessing the peritoneal tumor extension of gynecologic cancers [16].

Most tumors of the cervix are squamous cell carcinomas, known to be associated with exposure to human papillomavirus (HPV) and more frequent than adenocarcinomas of the cervix. While the diagnosis is biopsy-proven, the role of imaging is cancer staging. The International Federation of Obstetrics and Gynecology (FIGO) staging is essential for oncological therapeutic management. It includes carcinoma in situ (Tis), carcinoma confined to the uterus (T1), carcinoma invading beyond the uterus (T2), carcinoma extending to the pelvic wall and/or involving the lower third of the vagina (T3) and carcinoma invading the bladder or rectum (T4). Pelvic MRI is recommended for the local staging of cervical tumors as emphasized in the 2018 FIGO staging update [17].

In addition to the morphologic T2W sequences, DWI is used to assess the local extension of the carcinoma and is equivalent to contrast-enhanced MRI [18]. The T2W axial oblique plane perpendicular to the long axis of the cervix is important in assessing parametrial invasion (stage IIB) and can be coregistered with the high b-value DW sequence to improve tumoral tissue delineation [19], as demonstrated in Figure 2. Cervical carcinomas are characterized by hypercellularity resulting in high signal intensity (SI) on high b-value (1000 s/mm^2^) DW images and a low Signal Intensity (SI) on the ADC map compared to normal cervical stroma [16]. So far, no ADC cutoff value has been validated to predict the presence of malignancy, mainly because of the mutual dependence between the calculated ADC value and the range of b-values used for calculation [16]. In the context of follow-up after local radiotherapy and systemic chemotherapy treatment, DWI is used to differentiate between residual disease and local fibrosis [20], as well as to detect tumor recurrence [21]. DWI may also be used as a biomarker for monitoring tumor response [22,23]. In a recent meta-analysis on the use of artificial intelligence (AI) in gynecologic tumors, cervical cancer was subject to a high number of studies (34 from 71) mainly focusing on the prognostic value of imaging [24]. As all MR sequences are exploited collectively in AI, it remains difficult to extrapolate on the specific utility of DWI within this kind of black box approach.

Endometrial carcinoma is the most common gynecologic malignancy in developed countries, concerning women above 50-year-old. According to the Bokhman classification [25], Type I endometrial tumor is also known as endometrioid carcinoma is the most frequent type of cancer, with a generally favorable outcome. The 2nd most frequent histologic type of endometrial cancer corresponds to the papillary clear cell adenosquamous carcinoma and belongs to the type II group of tumors. Following FIGO classification, stage I of the tumor is limited to the body of the uterus and stage II is defined by an extension through the cervical stroma. In stage III the tumor locally invades the adnexa, the vagina or the parametrium and/or the pelvic floor, or presents para-aortic lymphadenopathy, whereas stage IV is defined by an extension of the tumor to the adjacent bladder or bowel or the presence of distant metastases.

MRI in endometrial cancer is performed for the staging of the disease. Invasion of less than 50% of the myometrium to separate stage Ia and Ib is based on a morphologic T2W plane perpendicular to the endometrial cavity. Endometrial cancer is usually hyperintense to the myometrium but can be difficult to differentiate from the surrounding tissue as illustrated in Figure 3. On DWI cancer shows diffusion restriction with a high b-1000 signal and low ADC values compared to the normal endometrium and adjacent myometrium. The addition of DWI to T2W imaging significantly improves the staging of endometrial cancer [26,27]. It is even more indispensable in patients with impaired renal function who cannot benefit from gadolinium administration, and therefore, from contrast-enhanced MRI. However, the combination of DWI and contrast-enhanced MRI remains the best approach to predict myometrial invasion, as supported by a recent study on machine learning [28]. DWI is also helpful in detecting other pelvic depositions in high grade tumors [8]. A false positive high signal on DWI with low ADC values in the endometrial cavity corresponds to secretory and hyperplastic endometrium or blood during the female cycle which is easily recognized by its high signal on T1W FatSat sequences [8].

Leiomyosarcomas are rare malignant tumors of the uterus and account for less than 10% of uterine cancers. The differentiation between benign leiomyoma and leiomyosarcoma is essential for the surgical management of these lesions. MRI and especially DWI play an important role in the characterization and management of both tumors. In addition to morphologic specificities of leiomyosarcoma, such as the intermediate T2 signal, nodular borders, hemorrhagic components, “T2 dark” areas and central unenhanced areas [29], DWI-based parameters constitute another essential tool to differentiate benign leiomyoma from leiomyosarcoma. As shown in Figure 4, uterine leiomyosarcoma usually shows low ADC values and increased signal intensity on high b-value DW images compared to the normal myometrium [15]. In the meta-analysis of Virarkar et al. which included 795 patients from eight studies, ADC values were significantly lower in leiomyosarcomas than in leiomyomas [30]. In a recent case-control retrospective study, Wahab et al. proposed a diagnostic algorithm to differentiate leiomyomas from uterine sarcomas based on the presence of lymphadenopathy, higher SI on high b-value images in the mass relative to the endometrium and ADC values inferior to 0.905 × 10^−3^ mm^2^/s [31]. The respective sensitivity and specificity of this algorithm to classify the uterine masses were 97% and 99% in a training set of 156 patients, 88% and 100% in a first validation set of 42 patients and 83% and 97% in a second validation set of 59 patients. Focally or globally reduced T2W SI and DWI-based SI lower than the endometrium allows to confidently diagnose the mass as benign [31]. However, this promising approach needs further validation by prospective multicentric studies.

Ovarian tumor is mainly an epithelial type of cancer (95%), including serous and mucinous cancers. The two other categories include the sex-cord stromal tumor and the germ cell tumor types. Ovarian cancer is the most lethal of all gynecologic cancers with the prognosis being determined by the initial staging at the time of detection. A precise characterization is, therefore, paramount to providing an accurate determination of the patient’s prognosis. The initial diagnosis is usually achieved by ultrasound examination while MRI is kept for indeterminate cases.

Normal ovaries usually show high SI on both the high b-value sequences and the corresponding ADC maps, corresponding to the so-called “T2 shine-through” effect. DWI is essential for the characterization of a suspicious solid component in heterogeneous complex ovarian masses, identifying solid high cellularity content in malignant ovarian tumors [32], as per the current European Society of Urogenital Radiology (ESUR) recommendations [33]. Illustrative MR images of adenocarcinoma can be found in Figure 5. The coregistration between the high b-value DWI and the morphologic T2W images is very efficient for this purpose. An adnexal lesion can be classified as benign when its solid component is hypointense on both the high b-value DWI and the T2W images (“dark/dark” lesion) [34]. However, DWI alone does not suffice to assess the malignancy of an ovarian tumor, as some benign lesions, such as mature cystic teratomas, endometriomas, or functional hemorrhagic cysts can show an impeded diffusion [16,32,35]. Dynamic contrast-enhanced MRI sequences are essential to further assess the probability of malignancy.

The important role of DWI in the characterization of ovarian tumors is well demonstrated in the recent introduction of the Ovarian-Adnexal Reporting and Data System (O-RADS)-MRI scoring system, an international effort to improve the standardization of adnexal MRI reports [36,37]. T2W images and DWI are sufficient to differentiate lesions with solid content in almost certainly benign cases (O-RADS-MRI 2) and higher (O-RADS-MRI 3 to 5), as the enhancement pattern of homogenously hypointense lesions on T2W and DW images does not impact the O-RADS-MRI classification [37]. The O-RADS-MRI risk score is built on a prospective, multicenter study in 1194 women with histologic examination and a 2-year follow-up imaging or clinical examination. The risk score yields an overall accuracy of 92%, a sensitivity of 93%, a specificity of 91%, a positive predictive value of 71% and a negative predictive value of 98% with a good agreement between junior and experienced readers, as attested by a kappa-score of 0.784 [36]. O-RADS-MRI validation and clinical acceptance are well advanced [38,39] and will be further improved when dedicated management recommendations are available [40].

Some pitfalls in the evaluation of diffusion-weighted images must be avoided. As mentioned previously, T2 shine-through, seen as a persistent hyperintensity throughout high b-value and ADC images, is one of them. Not all structures with high signal on diffusion are cancer and one must be aware that healthy tissues can yield low ADC values and high signal on high b-value images: normal endometrium, bowel, kidneys, spleen, lymph nodes [41,42]. Other criteria, such as the size, the heterogeneity and the very low ADC values can help to differentiate suspicious lymph nodes from normal ones. The normal endometrium in women of reproductive age can also show restricted diffusion because of the tissue’s high cell density. In this matter, quantitative evaluation of the tissue on ADC maps must be sought, as endometrial tumors present with even lower ADC values compared to normal adjacent tissue [15,16].

In conclusion, DWI is crucial to determine the malignancy of pelvic lesions and to assess their extension. It is an important sequence that must be part of all pelvic MRI examinations. Analysis of these sequences must use both the b-values sequences and the ADC map to avoid misinterpretation and must be compared to the signal of normal adjacent structure in the pelvis. It has to be analyzed in combination with the morphologic T2W, T1W and gadolinium-based sequences to avoid misdiagnosing some benign pelvic lesions as malignant.

## 4. Diffusion-Weighted Prostate Imaging

DWI represented a breakthrough in oncological imaging, including prostate imaging [43,44]. It now constitutes the cornerstone of prostate cancer imaging which is realized by multi-parametric MRI (mpMRI) [45]. The “multi-parametric” perspective corresponds to three sequences acquired in a single MRI session. It first includes the anatomical T2W sequence depicting the prostatic zonal architecture in at least two different planes, primarily in the axial and secondly in the sagittal and/or the coronal planes by using high resolution images. Then, two functional sequences are acquired in the axial plane. The considered DW sequence is performed prior to contrast administration and followed by DCE imaging, which is sometimes referred to as perfusion imaging. It has to be noted that the latter requires intravenous gadolinium contrast injection at a high flow rate. When no dynamic contrast-enhanced imaging is performed, the pair of T2W and DW sequences is referred to as bi-parametric MRI (bpMRI). BpMRI corresponds to the minimal standard of a contributive examination without loss in diagnostic performance, as illustrated in a study conducted on 431 patients [46].

Most of the prostate mpMRI examinations are based on ADC maps provided by only two different b-values and a monoexponential model where tissue diffusion is presumably free with Gaussian distributed distances [47]. However, more complex diffusion models have been developed to explore the microarchitectural tissue properties [48]. Increasing the amount of coefficient numbers in mathematical models improves the description of the water diffusion in the prostate as demonstrated by statistical methods, such as the Akaike Information Criterion [47,49,50]. For example, the addition of a second parameter in the biexponential model better illustrates the difference in diffusion properties due to multiple tissue compartments. The contribution of multiexponential models was investigated in the study of Bourne et al. using MR microimaging over prostatic formalin-fixed samples [49]. The voxel signal behavior first follows a rapid exponential decay and then, from b-values higher than 200 s/mm^2^, a slow exponential decay. Such observations led to the biexponential “three-coefficients” model, called the intravoxel incoherent motion (IVIM) model which takes into account the blood flow in the capillary network [51]. The IVIM model requires many b-values to produce three parametric maps of the perfusion fraction, the pure diffusion and the pseudodiffusion coefficient, respectively [47]. Pseudodiffusion originates from microscopic circulation in tissue predominant at low b-values (<200 s/mm^2^) and can be differentiated from the pure diffusion prevailing at higher b-values secondary to Brownian motion within the extravascular space [51]. Furthermore, the diffusion in prostatic tissue at b-values higher than 1000 s/mm^2^ is influenced by the spatial partitioning by cellular membranes which separate extracellular and intracellular spaces. This leads to another representation of a biexponential model with specific coefficients named ADCslow and ADChigh introduced for prostate DW imaging by Mulkern [52]. While multiexponential models add new terms for the DW signal decay, other models of higher complexity were studied in prostate MR imaging, such as the stretched exponential model or diffusion kurtosis imaging (DKI), which are not currently used in the clinical routine [53].

To this day, the monoexponential model keeps the high ground in routinely performed prostate mpMRI, due to its rapid processing and interpretation based on the ADC map. Such a monoparametric model is well integrated as a routine clinical standard [47]. Furthermore, its diagnostic performance was considered superior to the IVIM model [48,54]. In a recent study, the DKI model was not found superior to the monoexponential model for the detection of clinically significant cancers [55]. However, as suggested in the prostate imaging–reporting and data system (PI-RADS) guidelines, it is essential to interpret the ADC map in conjunction with the native DW images [45]. Figure 6 illustrates a neoplastic nodule arising in the peripheral zone of the prostate, appearing with lower ADC values in combination with a hyperintense focus on high b-values images. Such findings are typical for reduced diffusion inside a tumor area as a consequence of the high cellularity. Cellular density quantitatively measured by histological techniques was shown to be negatively correlated to ADC values [56,57]. Studies based on the comparison between axial DW images and histological slides have reported a negative correlation between ADC values and the differentiation grade of the tumor tissue—the Gleason score [58], which is more conspicuous in the peripheral zone than in the transition zone [59].

The essential role of the diffusion technique in prostate imaging is well illustrated by the PI-RADS score. This scoring system was established by a conjoined representation of the American College of Radiology (ACR), the European Society of Urogenital Radiology (ESUR) and the AdMeTech Foundation. The PI-RADS score ranges from 1 to 5, with normal MR findings (1) to MR findings highly suspicious of malignant prostate neoplasia (5). The 1.0 version was published in 2012 then followed by the 2.0 version in 2014 and the 2.1 version in 2019 [60,61,62]. Across versions 1.0 to 2.0, major changes revealed the role of diffusion imaging in prostate cancer detection. In version 1.0, all sequences performed in prostate MRI (T2W, DWI and DCE) were considered equal for cancer detection. Since PI-RADS version 2.0, DWI constitutes a key factor of appreciation. In the peripheral zone of the prostate, the PI-RADS score of tumor suspicion is primarily based on the diffusion findings as illustrated in Figure 6. In the transition zone, the parenchyma may be modified by stromal and glandular hyperplasia resulting in a complex and chaotic tissue that requires the anatomical information provided by T2W sequences. However, DWI is essential to complete the analysis, allowing to upgrade “equivocal areas” to higher PI-RADS score lesions. In a recent study, the performance of the PI-RADS score in prostate cancer detection was characterized by a high negative predictive value (NPV) of 94.1% for a PI-RADS cutoff at 3 and an NPV of 85.5% with a cutoff at 4 [63]. Prostate MRI examination provides a valuable tool for cancer risk stratification. However, a meta-analysis found the NPV of mpMRI to range from 63% to 98% illustrating the remaining limitations to detecting significant prostate cancer lesions in some patients [64,65]. The new version 2.1 of the PI-RADS score holds DWI as a reference for the detection of significant prostate cancer in the peripheral zone and as a highly valuable support tool for the transition zone next to the dominant T2W sequence.

In the PI-PRADS scoring system, the score’s assessment is commonly based on a qualitative and visual interpretation of the mpMRI. In such a frame, the radiologist’s experience and score reproducibility must be clarified. In a study of 2017 by Kwon et al., authors report an excellent interobserver agreement (k ≥ 0.870) in the DWI score and a good to excellent agreement (k ≥ 0.771) in the final PI-RADS score by using PI-RADS v2.0 between two radiologists of 11 years and 1 year experience, respectively [66]. However, previous studies reported lower interobserver agreement scores [66,67,68]. Therefore, in order to limit variability in mpMRI interpretation, optimization of MR protocol is the first and necessary step. The Prostate Imaging Quality (PI-QUAL) score [69] was specifically implemented for assessing the image quality of MR examinations, also including the assessment of DWI vulnerable to artifacts from rectal air. In addition, the learning curve and the length of the radiologist’s training have an improving effect on the inter-reader agreement [70,71]. The European Society of Urogenital Radiology (ESUR) and European Association of Urology (EAU) Section of Urologic Imaging (ESUI) consensus offered some recommendations suggesting radiologists be trained with a threshold of supervised cases before independent reporting, as well as a minimal yearly threshold of readings [70]. The expert panel and studies suggest a minimum of 100 training cases before obtaining an AUC on par with more experimented readers [70,71]. However, training requirements may be drastically modified by the introduction of new machine learning algorithms to assist prostate MRI analysis [72,73].

By incorporating DWI sequences into mpMRI of the prostate, the technique has evolved up to a dominant role in the clinical setting. Up to this point, clinical MR imaging of the prostate was limited to local staging prior to random bioptic sampling using trans-rectal ultrasound. MRI now holds more duties in the management of prostate cancer, including prostate cancer detection before biopsies, pre-operative staging, active surveillance of biopsy-proven low-grade cancer and detection of local recurrence after radical prostatectomy or other local therapy schemes. Siddiqui et al. demonstrated the importance of a prostate MRI examination before prostate biopsies to improve the detection of high-grade tumor lesions. Indeed, targeted biopsy based on MR images coregistered with ultrasound images during the procedure led to 30% more high-risk cancer detection in comparison with the standard procedure [74]. This finding translates into 20–30% of significant prostate cancer lesions being missed by standard systematic transrectal biopsies [75,76]. Furthermore, the role of prostate MRI in local staging was recently assessed by Caglic et al. [77]. According to them, mpMRI yields a sensitivity and specificity of 66.2% and 84.6% in extracapsular extension and 83.3% and 97.8% in seminal vesicle invasion, with comparable results using the biparametric MR examination. From this aspect, mpMRI provides vital local information by allowing the urologist to appreciate the surgical tactic for preserving the neurovascular bundles in Da-Vinci robot-assisted radical prostatectomy. After radical surgery, mpMRI can be used to analyze the operation situs in case of biochemical recurrence. However, the performance of the examination depends on Prostate Specific Antigen (PSA) levels and the Gleason grade of the initial neoplasia, as per the study of Venkatesan et al. which reported 87.2% negative examinations in cases of PSA less than 0.5 ng/mL in lower grade tumor and 88.9% positive examinations in cases of PSA higher than 1.5 ng/mL in higher grade tumor [78]. From this observation, the diagnostic contribution of mpMRI is most relevant in intermediate combinations of PSA levels and Gleason grade tumors.

In conclusion, DWI represents a major advancement in prostate cancer detection by MRI. MRI is recognized as a key factor in the actual scoring system PI-RADS 2.1, updated last in 2019. The quantitative ADC map derived from the acquisition of b-values images by fitting the monoexponential model is routinely used in the clinical setting and allows to obtain a representative tissue characterization. Low ADC values are correlated with high Gleason score tumor tissues composed of strong cellular areas. In the future, the role of DWI in bi-parametric MRI combining T2W and DW images without contrast administration must be more clearly defined as a potential method for better cancer risk stratification and for guiding prostate targeted biopsies.

## 5. Diffusion-Weighted Bladder Imaging

Bladder MRI in neoplastic disease is emerging while the local staging of urothelial carcinoma is essentially based on the cystoscopic findings and histological analysis of the tumor’s transurethral resection. The main criterion for the therapeutic decision is the integrity or invasion of the detrusor muscular layer, directing treatment towards radical surgery or more conservative endoscopic local resection. This duality leads to the major distinction between superficial, non-invasive muscle carcinoma and a musculo-invasive tumor. A score, the Vesical Imaging Reporting And Data System (VI-RADS) was introduced in 2018 [79] in addition to grades on a scale from 1 to 5. While the PI-RADS score in the prostate cancer mpMRI is dedicated to the probability of significant cancer, the VI-RADS score in bladder cancer grades the local staging of the tumoral lesion in the depth of the bladder wall. Each sequence of the mpMRI is assessed for the presence of a discontinuity in the bladder’s normal muscular layer, helping to assign a probability of muscular invasion between 1 (muscle invasion unlikely) and 5 (muscle invasion very likely). T2W images provide a first assessment because of their superior anatomical depiction of the bladder layers before DWI and DCE imaging contributes to the final category VI-RADS [79,80]. The scoring system’s performance was reviewed by Jury et al. [80] on the basis of six previous reports. Setting a cut-off at VI-RADS 4, a sensitivity and specificity range between 76–91% and 76–93%, respectively, with an interobserver agreement score over 0.7 [80]. It is clear that DWI is an essential part of the mpMRI in bladder cancer staging and further clinical validations of the VI-RADS score could call for an increasing contribution of the technique in routine practice.

## 6. Diffusion-Weighted Penile Imaging

In current practice, penile MRI is most frequently required by urologists for the local staging of tumors that mostly grow in the glans or prepuce, for penile fracture and for Peyronie’s disease. High resolution T2W images allow precise depiction of tumoral extension inside the corporea cavernosa or urethral wall, as well as traumatic tears or inflammatory thickening in the tunica albuginea of the corpora cavernosa. Under such well-defined clinical conditions, DWI has a less preponderant role. However, primary malignancies, most commonly squamous cell carcinomas, can be well identified using DWI and ADC maps. Penile neoplastic infiltration shows lower ADC values, in inverse proportion to the tumor’s histologic grade, according to the study of Barua et al. in 26 patients [81]. In cases of severe phimosis, tumors could be obscured [82] by the covering skin, making MRI examination particularly appropriate for a local exploration in search of neoplastic tissue before surgery.

## 7. Diffusion-Weighted Testicular Imaging

As for other organs, DWI is integrated into standard MRI protocols to characterize intratesticular masses, contributing to a scrotal multiparametric protocol similar to prostate imaging. Studies have investigated the role of DWI in the differentiation between non-germ cell tumors arising in the interstitial testicular tissue from germ cell neoplasms, or between seminomatous or non-seminomatous tumors [83,84]. Indeed, DWI provides functional information about the microstructural histological architecture which can combine with the microvascular tissue properties brought by DCE imaging in order to define the diagnosis. However, MRI only contributes in some specific cases because of the ultrasound’s high ability to demonstrate malignancies as vascularized and solid masses [85]. MRI may be performed when ultrasound findings are inconclusive in mass-like lesions, such as hematoma, segmental infarction, or granulomatous inflammation or infection [85,86]. With the support of the European Society of Urogenital Radiology (ESUR), recommendations are edited by the Scrotal and Penile Imaging Working Group on the appropriate indications of scrotal mpMRI [87]. In a promising perspective, studies have demonstrated a link between ADC and the testis spermatogenic function [88], although no clinical validation yet followed. Interestingly, a study by Ntorkou et al. on 49 men discussed ADC’s potential capability to predict a successful Microdissection Testicular Sperm Extraction for sperm retrieval in patients presenting nonobstructive azoospermia [89].

## 8. Diffusion-Weighted Renal Imaging

DWI has currently no established indication in the clinical diagnosis and management of kidney disease. However, multiple studies have reported promising observations supporting the possible role of DWI in various renal diseases. The technical description of the different acquisition or analysis methods specific to renal DWI is beyond the scope of this review and can be obtained elsewhere [90,91].

In oncology, ADC is not able to robustly differentiate between malignant and benign renal tumors but may be helpful to characterize tumor subtypes [92]. In a meta-analysis of 1126 renal lesions from 13 studies, clear cell renal cell carcinomas (RCC) demonstrated higher ADC values than non-clear cell RCCs, low-fat angiomyolipomas, papillary RCCs, and chromophobe RCCs, but lower ADC values than oncocytomas [93]. Although these results were recently confirmed using either ADC [92] or kurtosis tensor MR imaging—a more advanced type of DWI analysis [94]—the moderate sensitivity and specificity reported by these works may be insufficient for DWI to be a reliable single test to differentiate RCC subtypes [93]. In a study of 46 patients with Von Hippel–Lindau (VHL) disease, ADC at baseline was negatively correlated with the volume doubling time of 100 clear cells RCCs [95]. As active surveillance and kidney sparing surgery are important components of the management of VHL patients, assessment of tumor growth by ADC may be an important development of DWI in the future [96].

Acute pyelonephritis (APN) can be diagnosed by DWI with a reduction of the ADC value in the area of inflammatory cells infiltration [97,98,99]. Pyelonephritic foci appear dark on the ADC map. The DWI performance to detect APN is similar to both contrast enhanced CT (CECT) [100] and renal scintigraphy [101,102,103] and probably better than nuclear imaging for the detection of multiple inflammatory foci [103]. This could be advantageous in ambiguous situations where the diagnosis of APN is uncertain, especially in transplant kidneys and children. Therefore, DWI could replace traditional diagnostic tools without the need for iodine contrast media and sparing radiation doses. As emphasized by a recent report of an imaging task force of the European Society of Pediatric Radiology [104], DWI is currently not part of the current clinical guidelines and further studies are needed to better define the clinical role of DWI in the case of APN.

In acute graft dysfunction, ADC values are reduced in case of either acute rejection (AR), acute tubular necrosis (ATN), or immunosuppressive toxicity, but DWI is not able to differentiate between these pathologies [98,105,106,107,108,109,110]. An interesting application of DWI may be the selection of patients with acute transplant dysfunction who can benefit from a renal biopsy. In a retrospective study of 40 transplant kidneys, a combination of qualitative and quantitative DW-MRI parameters predicted the severity of histopathologic findings by comparison to normal or mild changes [111]. These results were confirmed in a prospective study of 33 transplant patients who needed an intervention [112]. DWI helped differentiate patients with or without the need for clinical management changes following a renal biopsy and may be an important step in a paradigm shift towards virtual biopsy [113].

Both cortical and medullary ADC values of diabetic kidneys are reduced by comparison to ADC values of well-functioning kidneys [114,115,116] and are correlated to the clinical stages of diabetic nephropathy [117]. However, the clinical utility of DWI for diabetic patients is not yet demonstrated and the results of large ongoing clinical trials, such as the Prognostic Imaging Biomarkers for Diabetic Kidney Disease (iBEAt) (ClinicalTrials.gov Identifier: NCT03716401) will help in this matter.

One of the most promising applications of DWI is the estimation of renal fibrosis in chronic kidney disease (CKD) patients, which is a key prognostic marker for renal function decrease and the progression of CKD. This assertion has two main arguments. First, ADC values correlate with renal function [118,119,120,121,122,123] and DWI is an accurate non-invasive imaging technique for the early diagnosis and staging of CKD, as shown by a meta-analysis [124]. Second, numerous clinical studies have directly demonstrated that cortical ADC values as well as other diffusion related parameters are linearly correlated to renal fibrosis in CKD patients, as assessed by renal biopsy [105,125,126,127,128,129,130,131,132,133]. As fibrosis increases, the cortical ADC values decrease much more than the medullary ADC values, resulting in an inversion of the cortico-medullary difference of ADC or so-called ΔADC which can be observed in the images [129] (see Figure 7). DWI can also differentiate between the different levels of fibrosis [127,128].

The exact physiological and mechanistic explanation for the link between DWI values and renal fibrosis is not fully understood but may be the result of two main synergetic events occurring in the development of renal fibrosis in CKD: (1) an increase in cellular density and the extra-cellular matrix which reduces free water motion and (2) a reduction of microvascular perfusion and filtration-induced water mobility expected with impaired renal function [134].

Interestingly, the correlation of DWI with renal fibrosis is independent of the renal function as measured by eGFR, further supporting the role of DWI as a surrogate of renal fibrosis [127]. Whether DWI may allow reducing the number of renal biopsies by predicting the amount of interstitial fibrosis in CKD is currently not known and needs further evaluation.

Another potential application of DWI in CKD is related to the prediction of its evolution. Recently, three independent research groups observed that baseline DWI was correlated to the decline of renal function [126,135,136]. In a study with 197 CKD patients, ΔADC prediction of the worst renal outcome as shown in Figure 8 was independent of baseline age, sex, eGFR and proteinuria, strongly suggesting that DWI could be an independent prognostic marker of CKD.

In conclusion, DWI shows strong promise to characterize renal tumors, detecting APN and quantifying renal fibrosis in diabetic and CKD patients. With the ongoing clinical trials, as well as international efforts for methodological uniformization [137], its diagnostic and predictive power is likely to be significantly improved. The use of DWI may be further amplified by the development of multiparametric MRI allowing simultaneous additional assessment of renal blood flow and oxygenation as well as other morphometric parameters, such as the relaxation time T1 on top of DWI [128].

## 9. Study Limitations

This review is based on a literature analysis performed using the PubMed database. For each genitourinary organ, the most relevant studies in terms of recency and representative impact in the field of DWI were selected. Queries were performed by searching the database for keywords regarding the involved organ and involving diffusion imaging and/or MRI. However, this work does not report studies in the way a structured meta-analysis does by including scientific articles meeting systematic criteria. The selected works’ pertinence was established from the viewpoint of our actual practice and from the MR examinations routinely performed at our institution. In such a setting, the study selection contains an acknowledged part of subjectivity mostly relying on the authors’ combined experience.

## 10. Conclusions

DWI is already a well-established imaging technique for the assessment of the genitourinary system with the potential for significant improvement. In the female pelvis cancer and prostate cancer, DWI is part of the international guideline for the diagnosis, staging and assessment of recurrence. DWI is even a determinant in the standardized risk scores O-RADS and PI-RADS. In the kidney, the most promising applications of DWI may be the quantification of renal fibrosis and the prediction of disease evolution in CKD. In order to further integrate DWI into the diagnosis strategies and patient management, standardizing the acquisition and analysis protocols will be the next challenge.

## Figures and Tables

**Figure 1 jcm-11-01921-f001:**
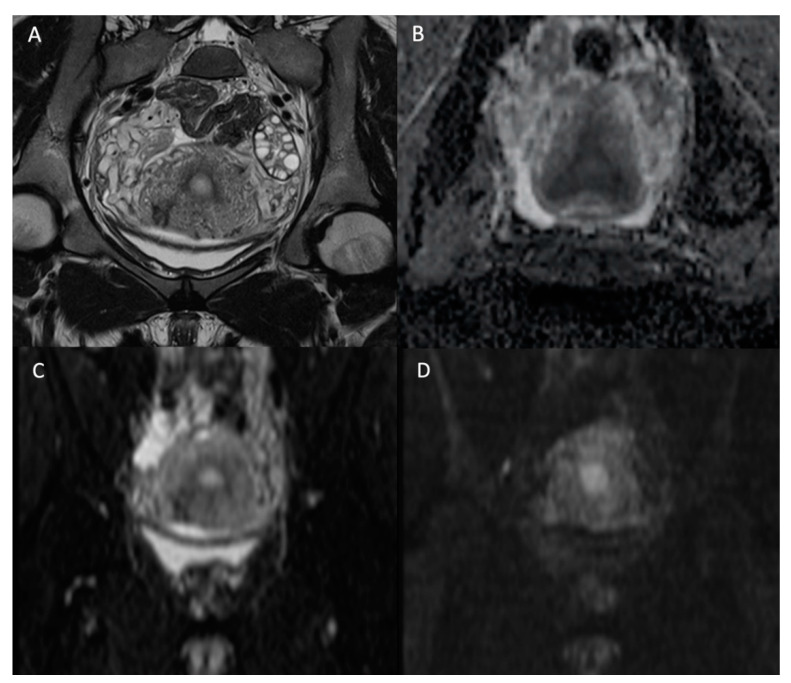
Normal female pelvis of 26-year-old in the coronal plane. (**A**) T2W image; (**B**) ADC map; (**C**) b-value = 0 s/mm^2^ DW image; (**D**) b-value = 1000 s/mm^2^ DW image. We see the disappearance of high fluid signal (as the one in the bladder) with increasing b-values but persistence of high signal intensity on high b-value for the endometrium.

**Figure 2 jcm-11-01921-f002:**
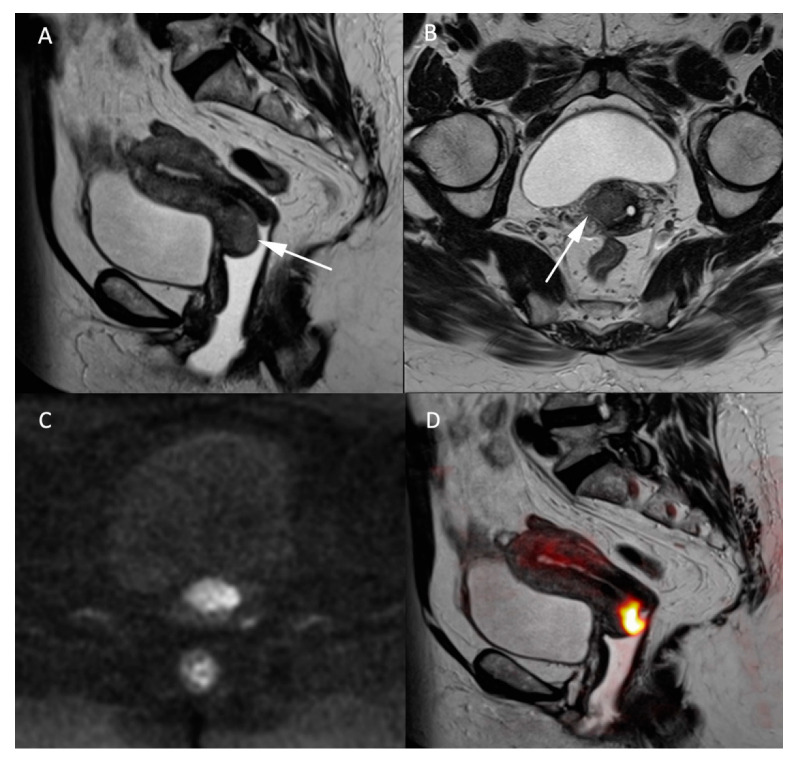
MR images of a 66-year-old woman with a known cervical carcinoma. (**A**) Sagittal T2W image; (**B**) axial T2W image perpendicular to the cervical axis. Cervical cancer and its extension appearing as low-contrasted T2W area (arrow) through the normal stroma and the right parametrium, (**C**) high b-value (b = 1000 s/mm^2^) and (**D**) fusion images between T2W and high b-value sequences for better evaluation of the carcinoma’s extension.

**Figure 3 jcm-11-01921-f003:**
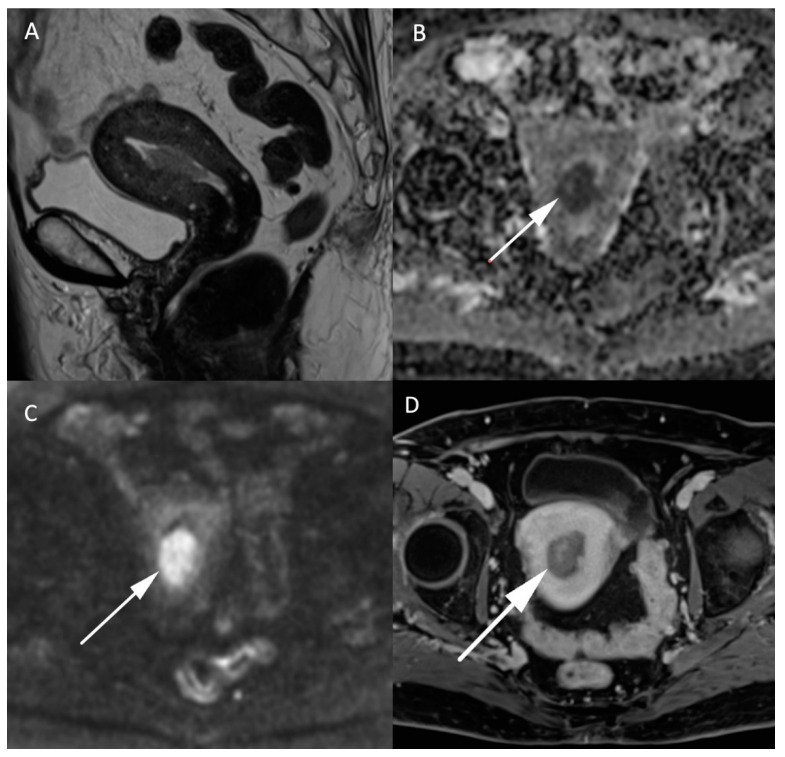
MR images of an endometrial carcinoma in a 93-year-old woman. (**A**) Sagittal T2W image in the endometrial cavity with extension in the myometrium smaller than 50% of its thickness. (**B**) ADC map shows a restricted diffusion in the endometrial carcinoma visible as a “dark” area (arrow) in opposition with (**C**) high signal (arrow) on high b-value images (b = 1000 s/mm^2^). (**D**) post injection of gadolinium T1W image shows the endometrial carcinoma (arrow) with an enhancement less than the myometrium’s muscle.

**Figure 4 jcm-11-01921-f004:**
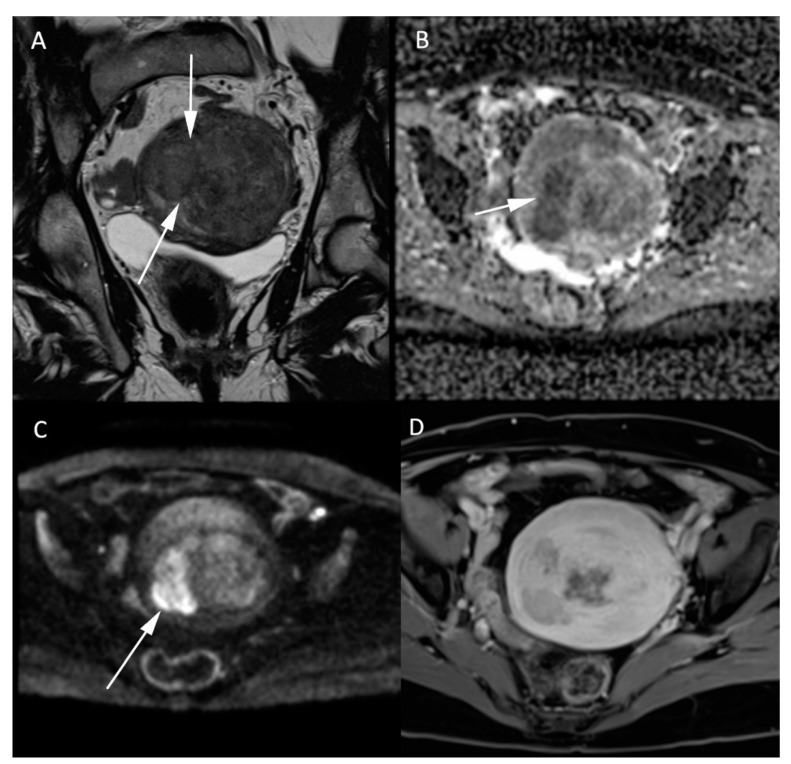
MR images of a leiomyosarcoma in a 54-year-old woman. (**A**) voluminous leiomyosarcoma with an intermediary T2W signal and irregular borders (arrow). Part of the leiomyosarcoma demonstrates a diffusion restriction with low (**B**) ADC values and high signal on the (**C**) b-1000 sequence. (**D**) post injection of gadolinium T1W sequence shows the absence of central enhancement consistent with central necrosis. All features are characteristic of malignancy within a leiomyoma.

**Figure 5 jcm-11-01921-f005:**
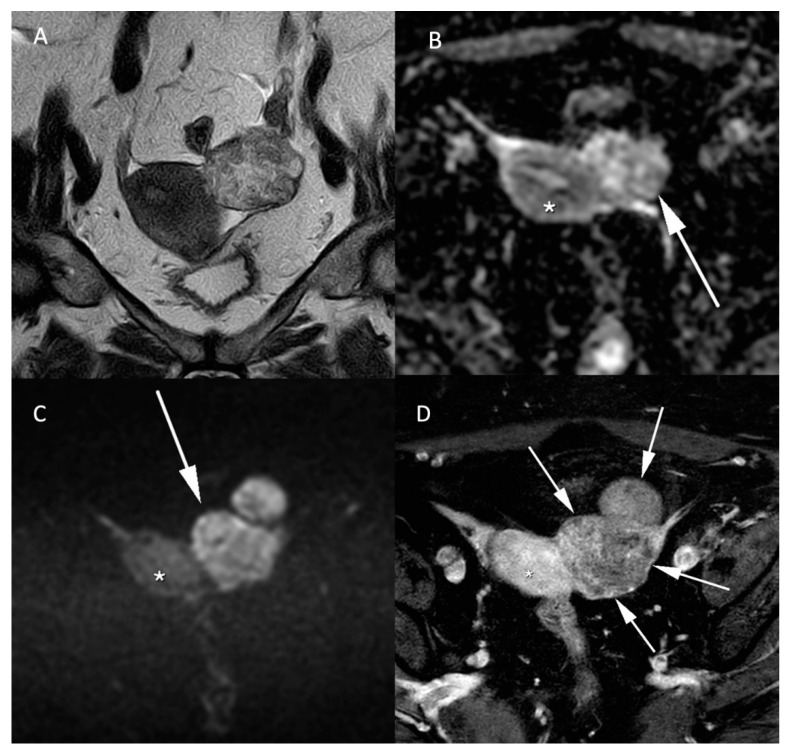
Histologically proven left ovary adenocarcinoma in a 64-year-old woman. (**A**) T2W hyperintense heterogeneous left adnexal mass next to the uterus (*). Tissular bilobed left adnexal mass with parts of low (**B**) ADC values and high (**C**) b-1000 signal consistent with a diffusion restriction in the lesion (**C**). Post injection of gadolinium (**D**) T1W sequence with fat-saturation shows a heterogeneous enhancement (arrow).

**Figure 6 jcm-11-01921-f006:**
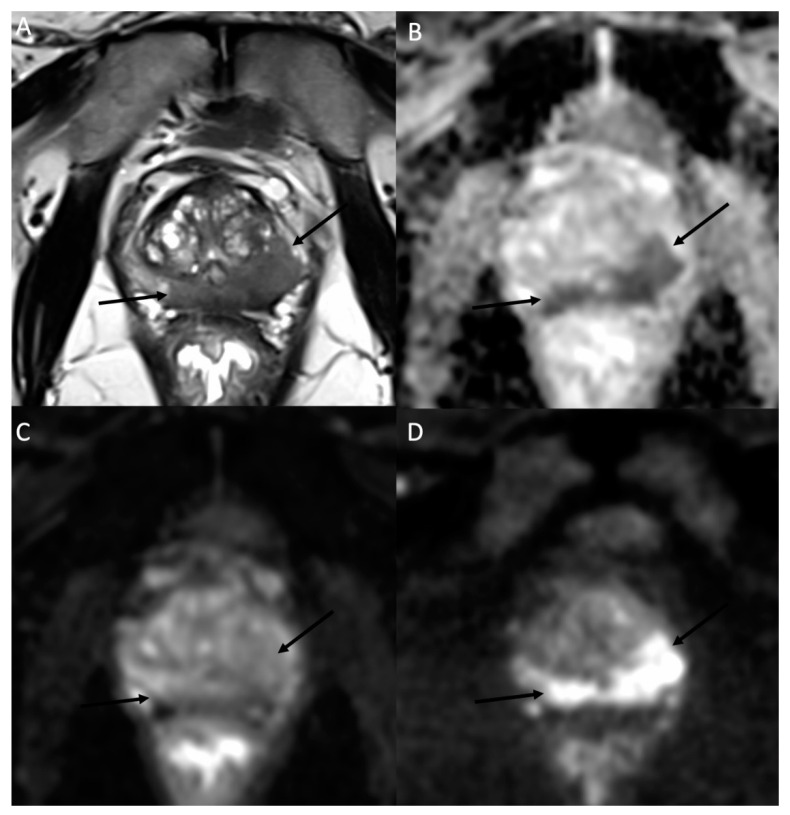
MR images of a prostate MRI at 3T in a 78-year-old man with a 7.7 ng/mL PSA. (**A**) The axial T2W morphologic image shows a hypointense area (arrows) expanding in the peripheral zone of the prostatic parenchyma on both sides with a predominant tissular infiltration on the left side. (**B**) The corresponding ADC map obtained by fitting a monoexponential model using the b-50 (**C**) and b-1500 (**D**) images. As a typical example of high-grade prostate adenocarcinoma, the b-50 (**C**) to b-1500 (**D**) images show an increasing signal in the neoplastic tissue due to the diffusion restriction properties in histology proven Gleason 9 prostate carcinoma. These MR findings were reported as a PI-RADS 5.

**Figure 7 jcm-11-01921-f007:**
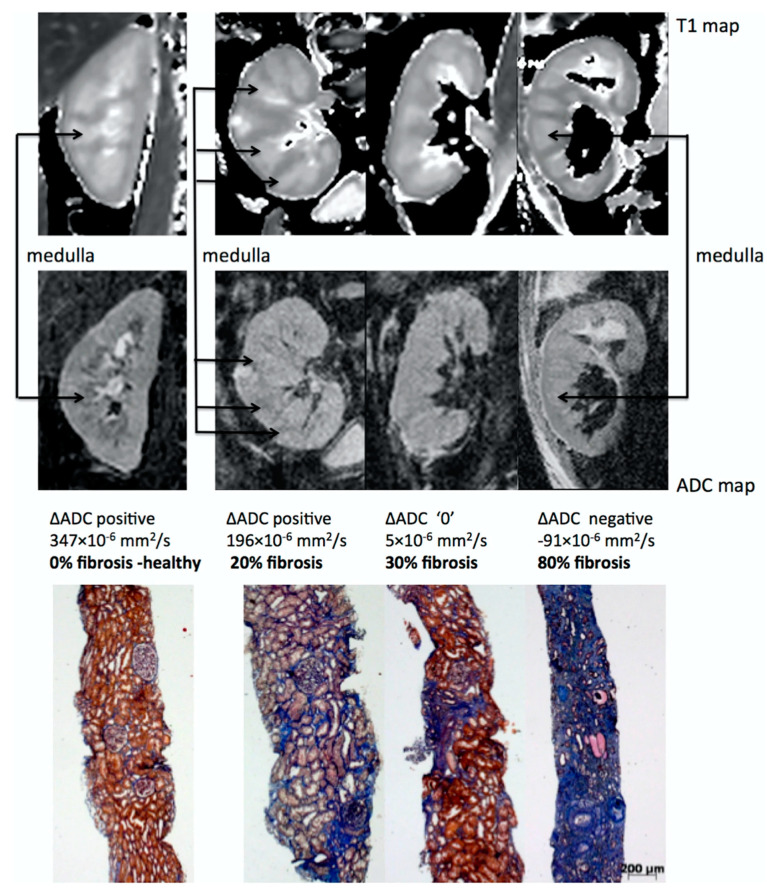
Representative biopsy and MR images in chronic kidney disease patients. Morphological MOLLI T1 maps used for the positioning of the regions of interest (top row) and ADC maps (lower row) for three patients showing different ΔADC cases: positive, zero and negative; along with the corresponding fibrosis levels from histology (Masson trichrome staining). The inversion of the corticomedullary ADC difference meets the increasing degree of renal fibrosis. Adapted from Figure 7 of Ref. [129] with permission. Copyright 2016 Springer Nature.

**Figure 8 jcm-11-01921-f008:**
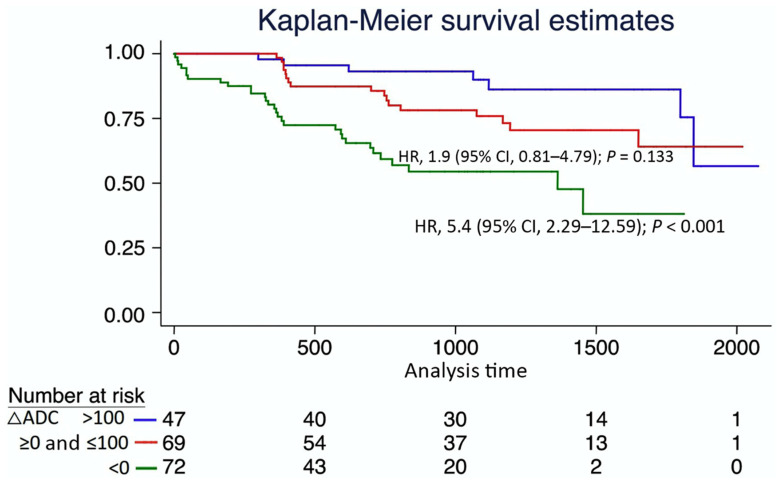
Kaplan-Meier survival curves, stratified according to the corticomedullary difference of the apparent diffusion coefficient ΔADC in chronic kidney disease patients. The primary outcome was a decline of eGFR > 30% or renal replacement therapy. A negative ΔADC was associated with a rapid decline in renal function. CI, confidence interval; HR, hazard ratio. Adapted from Figure 2 of Ref. [126] with permission. Copyright 2016 Springer Nature.

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
