# Peer review of "Diffusion-Weighted MRI in the Genitourinary System"

_jcm, 2022, doi:10.3390/jcm11071921_

Round 1

Reviewer 1 Report

I read with interest this nicely written literature review regarding the role of DW MRI in diagnosis of genitourinary tract. I believe that this manuscript can be potentially very interesting and give an overview / summary of the current literature. However, I have some concerns:

  • Most of the manuscript is about female pelvis, while the other part of the manuscript is about Prostate and Kidney. From my point of view there are some organs completely missed. What about Bladder and other organs (penis and testicles?)? I would reduce the female pelvis session and insert a small session for these organs even if the DW MRI is less used. 
  • I would introduce some info regarding the training / learning curve in order to correctly interpret the MRI and about the interobserver agreement in the different organs.
  • Even if it's a non-systematic literature review, I would introduce a short paragraph with methods, after the introduction, to clear how you performed the literature search and the writing.
  • Similarly, some limitations must be declared

Reviewer 2 Report

The authors describe very useful and comprehensive article about diffusion -weighted MRI. The article is very well designed and easy to read. The statistical analysis is adequate. 

The authors compose very comprehensive and skilful  article about the subject. Actually,  it is a very concise work on such difficult subject. I like the most step by step  following subject of diffusion-weighted MRI in the genitourinary system.  I enjoyed much reading it. I was particularly impressed by authors honesty about today's proven and research still in progress.  Judging from my point of interest, the paper adresses various subjects of genitourianry system  giving cpmprehensive information on particular themes. The main quality is that one article contains all recent necessary data in one place. The paper is somehow not easy to read to the unskilful reader to the subject,  and requires some degree of prior konwledge. If extended to be easy to read to all potential readers, the content should be much extended and therefore unsuitable to the connoisseaurs.  I believe that the paper containes all necessaty data for potential readers.

Round 2

Reviewer 1 Report

Dear authors,

I carefully read your revised manuscript and I believe it is much improved. Thank you for having addressed my concerns.